# The influence of social networks in adoption of integrated health interventions: A qualitative study of fishermen in Malawi

Madalo Mukoka[1,2,3]*, Owen Mhango[1,3], Hussein H. Twabi[1,4], Chikumbutso Chipandwe[1], Robina Semphere[1,5], Takondwa C. Msosa[1,6], Marriott Nliwasa[1,5], Chisomo Msefula[1], Guy Harling[7,8,9,10], Alison Price[11,12], Katherine Fielding[2], Augustine T. Choko[3,13], Moses K. Kumwenda[1,3,14]

1 Helse Nord Tuberculosis Initiative, Department of Pathology, Kamuzu University of Health Sciences, Blantyre, Malawi, 2 Department of Infectious Disease Epidemiology and International Health, London School of Hygiene and Tropical Medicine, London, United Kingdom, 3 Public Health group, Malawi-Liverpool-Wellcome Programme, Blantyre, Malawi, 4 Institute of Life Course and Medical Sciences, University of Liverpool, Liverpool, United Kingdom, 5 School of Health and Wellbeing, University of Glasgow, Scotland, United Kingdom, 6 Department of Global Health, Amsterdam University Medical Centres, Amsterdam, the Netherlands, 7 Institute for Global Health, University College London, London, United Kingdom, 8 Africa Health Research Institute, KwaZulu-Natal, Durban, South Africa, 9 MRC/Wits Rural Public Health & Health Transitions Research Unit (Agincourt), University of the Witwatersrand, Johannesburg, South Africa, 10 School of Nursing & Public Health, University of KwaZulu-Natal, Durban, South Africa, 11 Malawi Epidemiology and Intervention Research Unit, Lilongwe, Malawi, 12 Department of Population Health, London School of Hygiene and Tropical Medicine, London, United Kingdom, 13 Department of International Public Health, Liverpool School of Tropical Medicine, Liverpool, United Kingdom, 14 Institute of Resilient Health Systems, Department of Clinical Sciences and Department of International Public Health, Liverpool School of Tropical Medicine, Liverpool, United Kingdom

* madalo.mukoka@gmail.com, mmukoka@kuhes.ac.mw, Madalo.Mukoka@lshtm.ac.uk

## Abstract

### Introduction

Social networks play a vital role in influencing individual and collective health behaviors and facilitating the diffusion of health interventions. Fishing communities in sub-Saharan Africa face socio-cultural and occupational challenges such as low literacy rates and high mobility. However, their high social cohesion creates a unique social environment for spreading health interventions. We explored the role of social relationships in mediating health intervention adoption in these communities, to inform future intervention strategies.

### Methods

We conducted an exploratory phenomenological qualitative study between September and October 2024 nested within a large community cluster randomized trial (the FISH trial) in fishing communities in Mangochi district on the southern tip of Lake Malawi. We conducted four focus group discussions (N = 47) and 16 in-depth

**Data availability statement:** The dataset supporting the conclusions of this article are available via the London School of Hygiene & Tropical Medicine Data Compass https://data-compass.lshtm.ac.uk/

**Funding:** Research reported in this publication was supported by MN's Helse Nord RHF grant (2019/995) and in part by ATC's Wellcome Trust (grant number 216458/Z/19/Z) and the UK National Institute for Health Research grant (grant number 216458/Z/19/Z). G.H. is supported by a fellowship from the Wellcome Trust and Royal Society (grant number 210479/Z/18/Z). The funders had no role in study design, data analysis, decision to publish or preparation of the manuscript.

**Competing interests:** The authors have declared that no competing interests exist.

interviews. Data were analyzed thematically using a hybrid deductive and inductive data coding approach.

## Results

Fishermen's social networks included fishing friendships, kin and broader community connections that facilitated knowledge exchange, resource sharing and support provision. Familiarity, trust, desire to conform to community norms and shared community ties enabled health knowledge exchange and encouraged uptake of targeted interventions for HIV and schistosomiasis, despite fishermen's mobility. Other facilitators at individual level included perceived susceptibility and financial incentives. Occasionally, negative rumors spread through peer networks, e.g., a link between blood sample collection and malign government agenda, contributed to disengagement with health interventions. Other factors including HIV-related stigma and prohibitive traditional beliefs interacted with social networks to hinder uptake.

## Discussion

Fishermen's health decisions are deeply embedded within their social structures. However, other factors operating at the individual and community level were noted to be crucial catalysts of the decision-making process. This study highlights the potential of leveraging social networks in public health strategies for mobile communities so long as they account for critical individual factors that influence health service engagement.

## Introduction

Social networks are defined as the relationships or connections that exist between various individuals, linked by different forms of interdependence, including friendship, kinship, financial exchange, and other social or professional interactions [1]. The influence of social networks on health behaviors has gained substantial attention in public health [2–5]. In the context of health interventions, social networks remain an under-exploited potential resource that can be leveraged to shape beliefs, facilitate or hinder information flow, or facilitate access to resources and thus accelerate uptake [6,7].

Social networks critically foster knowledge exchange and behavior change in contexts where formal health information may be less accessible [1,8]. This is especially relevant in rural sub-Saharan communities, where low literacy rates, high mobility, and occupation-specific challenges intersect with high social cohesion, creating a unique social environment for health intervention diffusion [9,10]. Fishing-focused communities, such as those in Mangochi, Malawi, face a distinct set of socio-cultural challenges, leading to higher rates of communicable diseases, such as HIV, Schistosomiasis, and Malaria, often exacerbated by limited access to healthcare services [11–13]. The vulnerability to communicable diseases is particularly pronounced among fishermen who are young, highly mobile, use alcohol heavily, and usually

embrace exaggerated forms of masculinity in response to their social marginalization [9,14]. Gender-related studies reveal that masculine norms, including risk-taking, emotional control, dominance, primacy of work and self-reliance hinder men, including fishermen, from engaging in health services - further worsening their vulnerability to communicable diseases and poor health outcomes [15–19].

Public health interventions in low resource settings sometimes rely on informal social structures and networks as a cost-effective medium for both disseminating and reinforcing health-related messages [7,20]. Given the close-knit nature of fishing communities and the high level of mutual reliance among fishermen, understanding how social networks influence health behavior and intervention uptake is crucial to developing effective, culturally appropriate health strategies in these settings [21,22]. However, there is some evidence that the evolution of the fishing trade and changes to the political and economic environment may have eroded the social connectedness that was once shared in fishing communities, which may negatively affect information dissemination required in health interventions [23,24].

While previous research has documented the role of social networks on health behavior in various contexts, there is limited literature focusing specifically on relatively cohesive and isolated communities common in rural sub-Saharan Africa [25–27]. To address this gap, this study was nested in the FISH trial (Creating demand for Fishermen's Schistosomiasis and HIV services), a 3-arm cluster randomized trial (CRT), designed to jointly deliver schistosomiasis and HIV services to fishermen in Mangochi, Malawi [28,29].

This study, through a qualitative inquiry, sought to investigate the direct and indirect influences of social relationships on uptake of integrated health interventions among fishermen in Mangochi, Malawi. The study focused on the social relationships that were important to the fishermen, dynamics of information exchange, social influence, and other factors influencing intervention adoption, with the goal of informing future intervention strategies tailored to the unique characteristics of fishing communities in this region.

## Methods

### Setting

The study was conducted in Mangochi district in the eastern region of Malawi and is mainly agrarian with fishing activities along the shores of the lake [30]. The study was conducted within the catchment areas of FISH trial and among fishermen residing at four landing sites. A landing site, a place where fishermen leave their boats, consisted of multiple boat crews (approximately 10 crews per site, with each boat crew having an average of 10 members). A landing site was also referred to as a cluster in the FISH trial.

**The FISH trial and sampling.** The FISH trial, conducted from March 2020 to January 2023, compared three strategies (peer educators (PE) or peer-distributor-educators (PDE) vs beach-side services alone (standard of care [SoC]) for creating demand for services among fishermen. Two of the strategies in the trial (PE and PDE) used peer-nomination of influential fishermen to deliver information on HIV and schistosomiasis, to distribute invitation coupons, and to provide HIV self-testing kits (in the PDE arm) to fellow fishermen in their boats [29]. The integrated schistosomiasis and HIV services provision was through a beach clinic erected in the cluster. The study found strong evidence for the increase in the uptake of study interventions and that the added distribution of HIV self-test kits through peers promoted high engagement with study services and reduced the prevalence of active schistosomiasis [28].

To examine the influence of social relationships on intervention adoption, we focused exclusively on sites that were allocated to the intervention arms of FISH trial (i.e., PE or PDE arms). For this work, four clusters were selected from a pool of 30 (i.e., 15 clusters that recruited participants into the PE arm and 15 into the PDE arm) – Fig 1. We selected two best performing and two poorly performing clusters based on the trial outcome of uptake of HIV testing. In the end, we included two best performing clusters, F001 and F045, that recruited participants to the PE and PDE arm respectively and two poorly performing clusters, F024 and F043, that also recruited participants to the PE and PDE arm respectively.

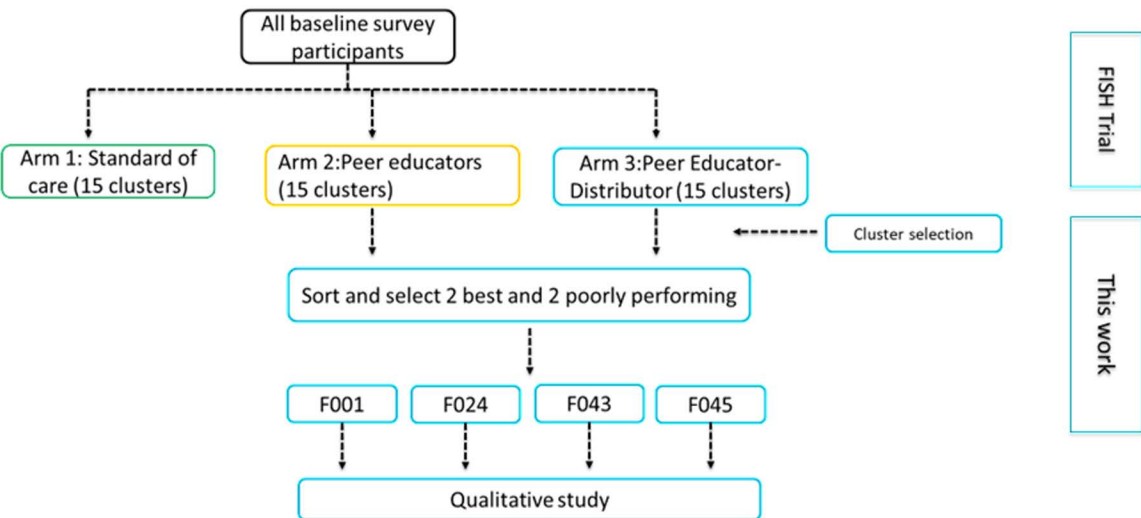

**Fig 1. FISH trial arms, cluster selection, and activities.**

## Conceptual framework

A conceptual framework based on Social Network theory was developed and guided the design and development of data collection tools- Fig 2 [31]. The framework posits that social influence, mediated by knowledge, social network interactions, and access to resources, affects intervention adoption. The socioecological framework was used to account for other factors operating at individual and community level [32].

## Study population and recruitment

The data for this study was collected between 3rd September and 11th October 2024. We conducted four focus group discussions (FGDs) and in-depth interviews (IDIs) with purposively selected fishermen. The FGDs included 47 male participants (each FGD consisting of 11–12 individuals from a single cluster) who were at least 18 years of age. We selected both fishermen who participated in the FISH trial and those who had not. For those that participated in the trial, we purposively selected individuals based on age (facilitating the inclusion of an equal number of participants aged ≤30 years, as well as>30 years). Fishermen naïve to the FISH trial were recruited using convenience sampling. The IDIs included 16 fishermen (four participants per cluster) all of whom had participated in the FISH trial. The IDI participants were different from the FGD participants. The relationship between the researchers and the participants who had previously participated in the FISH trial was established during the implementation of the trial. However, there was no prior relationship between researchers and the FISH trial naive fishermen. The interviews were conducted at a convenient location identified by the cluster peer educators such as the beach, under trees or participants' verandas. They were agreed upon by all the participants (independently in the case of IDIs). Peer educators were members of the fishing community and peer-nominated to support FISH trial activities.

For the FGDs, we recruited individuals that were readily available on the day through the peer educator with their details being cross-checked against the FISH trial data. We did not encounter any refusals, perhaps due to the good relationship that the research assistants (who also supported FISH trial work) had developed with the participants in the past and prior community engagement activities.

Semi-structured guides, informed by the study objectives and conceptual framework, were developed for this study. Both the IDI and FGD guides were back-translated, pilot-tested and refined in another cluster that was not included for

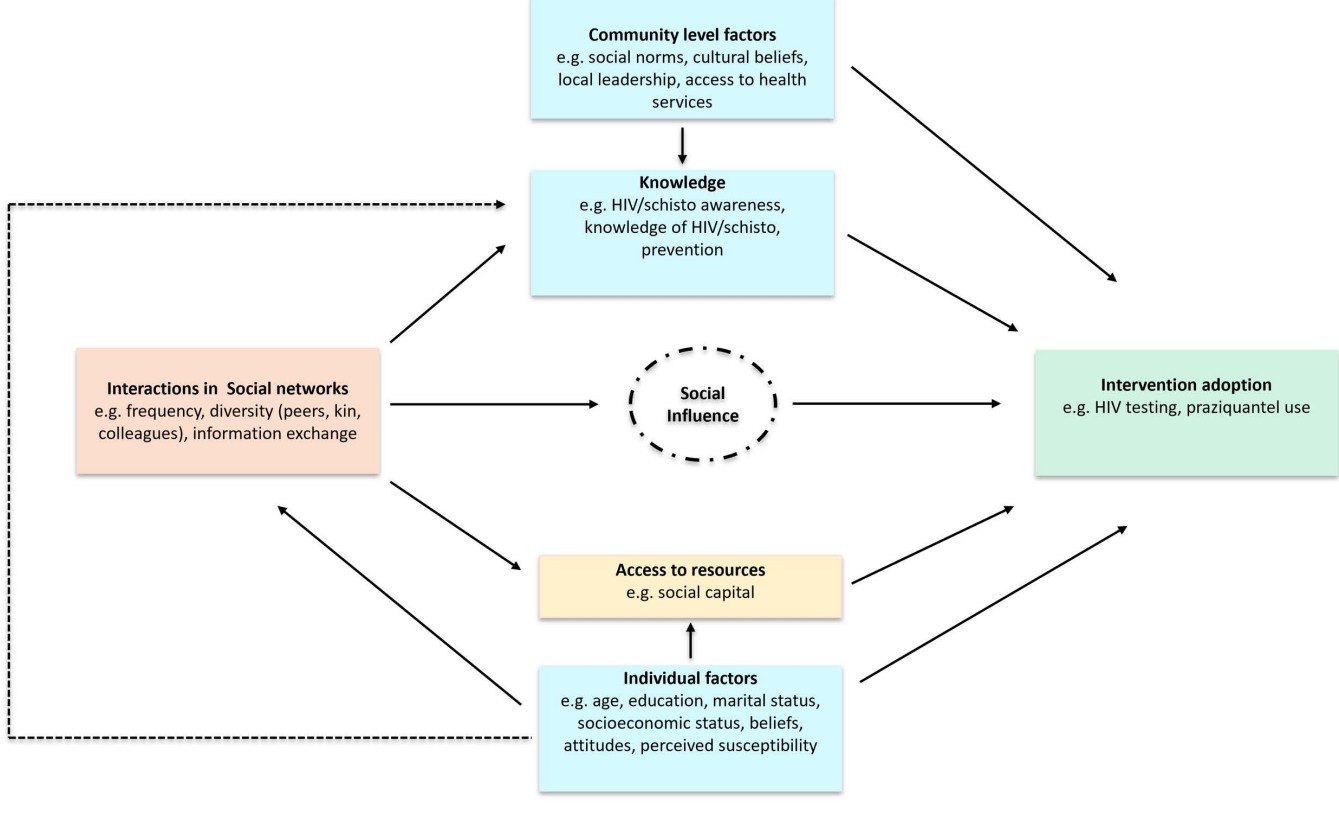

**Fig 2. Conceptual framework.**

participation in this study. The interviews were conducted face-to-face, in Chichewa (local language commonly used in the area) by a team led by MM (MSc) and OM (BSoc), both trained in qualitative methods. Collectively, the team of field researchers had five years' experience in conducting qualitative research. Specific to the current study, members of the research teams received protocol training that included qualitative data collection methods. The six field researchers (5 males and 1 female: MM, OM, RM, KP, KM and WS) used open-ended questions and probes aiming to explore key issues: the influence of social relationships in the uptake of interventions; mechanism of intervention adoption in the FISH trial; norms around HIV and schistosomiasis service provision; and barriers and facilitators to HIV and schistosomiasis testing and treatment. The same themes were addressed in the IDIs, but with a stronger emphasis on gathering individual perspectives. Repeat IDIs were conducted in two clusters to further investigate gaps in concepts around mobility and misinformation.

IDIs lasted an average of one hour and the FGDs were on average two hours long. The interviews were recorded using digital audio recorders and field notes containing summaries for each interview were prepared. Daily briefing meetings were held with MK (Senior Behavioral Scientist) to discuss emerging concepts or themes and reflect on the influence of the researchers during the data collection process. This reflexive and iterative process helped to improve the data collection process and enrich the collected data during fieldwork. Observable through these briefing meetings, there was evidence of data saturation as recurring responses and consistent perspectives were apparent across participants. Transcripts were not returned to study participants for comments or clarifications.

## Ethical considerations

The study was approved by Kamuzu University of Health Sciences Research and Ethics Committee (COMREC), approval number: P.05/24-0799 and London School of Hygiene and Tropical Medicine (LSHTM) Ethics committee, approval number: 31119. All participants gave a written informed consent in the local language or a witnessed consent plus thumb-print if illiterate before participation. An impartial witness, identified by the participant, read and helped the participants to understand the study information before consent was given. To maintain anonymity, participants were assigned an identification number during the FGDs, IDIs, and throughout the data analysis process.

## Data analysis

The recorded data was transcribed and translated by OM, RM, KP, KM and WS who were adequately experienced and trained to perform this function. All the transcripts underwent quality checks, for both accuracy and intelligibility, by two independent researchers (FK and WL). A codebook was generated using a hybrid approach- both deductive and inductive. Coding was performed by MM, OM and CC who received prior training using NVIVO by MK. The coding framework was informed by the conceptual framework, objectives and topic guides. Firstly, data familiarization was done, allowing for an initial understanding of the content and the identification of potential patterns. The initial coding was primarily deductive, guided by the conceptual framework and objectives before being imported into NVIVO (version 14) [33].

Parallel coding for each transcript was done and through this process, the priori codes initially incorporated in the coding framework were refined and expanded iteratively through an inductive process. This coding process involved reading and rereading each transcript, generating codes, team discussions, and comparison of emergent codes to ensure consistency. A total of 132 codes were generated in the process, however, not all of them were directly relevant to the focus of this write-up.

The next stage involved theme generation, which involved grouping and regrouping codes into broad thematic categories and then attaching theme labels to each distinct category. Through this process, the final codebook was structured hierarchically, with broad thematic categories at the top level and sub-themes nested beneath. The focus was on identifying themes that captured the essence of the participants' experiences and addressed the research objectives. The senior behavioral scientist (MK) played a critical role in this process by providing oversight and guidance throughout the coding and theme development stages. He facilitated coders' training, discussions among coders to resolve discrepancies and provided methodological expertise in the transition from coding to theme generation. Finally, participants were provided with feedback on the findings.

## Findings

### Participation and participant characteristics

Of the 47 FGD participants, 30 (64%) had previously participated in the FISH trial. Each FGD had 12 participants except in cluster F024 where there were 11- Table 1. We interviewed four participants for IDIs per cluster as planned.

We present the findings under three key themes that emerged: perceived significant social ties for fishermen; networks' influence on behavior and intervention uptake; and other barriers and facilitators operating at the structural, community, individual and economic level affecting uptake of health interventions. Table 2 provides the summary of the key findings and representative quotes.

### Perceived significant social ties for fishermen

**Fishing team ties.** Participants narrated that fishermen belonged to boat teams of 10 or 11 locally known as 'Kampani' (Company). Each team was well structured with members performing defined roles and responsibilities. Having a cordial

**Table 1. Characteristics of participants by interview type.**

|  | FGD | IDI |
|---|---|---|
| n | 47 | 16 |
| Age (mean [SD]) | 35.30 (13.27) | 41.19 (13.70) |
| Cluster ID (n [%]) |  |  |
| F001 | 12 (26) | 4 (25) |
| F024 | 11 (23) | 4 (25) |
| F043 | 12 (26) | 4 (25) |
| F045 | 12 (26) | 4 (25) |
| Education (n [%]) |  |  |
| No formal education | 8 (17) | 2 (13) |
| Primary | 22 (47) | 9 (56) |
| Secondary | 17 (36) | 5 (31) |
| Tertiary | 0 (0) | 0 (0) |
| Marital Status (n [%]) |  |  |
| Never Married | 13 (28) | 2 (13) |
| Married | 33 (70) | 14 (88) |
| Divorced | 1 (2) | 0 (0) |
| FISH trial participation (n [%]) |  |  |
| Participant | 30 (64) | 16 (100) |
| Non-participant | 17 (36) | 0 (0) |

relationship with other team members was valued and determined one's future in the team and these sentiments were shared across the four distinct and geographically separate landing sites.

'… Mostly, they look for fishermen with good behavior because one has to relate well with the fishermen whom he works with…This makes one to be in good terms with his friends within the boat company.'- FGD particiant_F001_P11_22 years old.

These teams established important social bonds because they spent long hours offshore and in cases when they were away from home for prolonged times in search of fish. These ties were recognized as being an essential part of a fisherman's life. These relationships and social bonds were valuable for sharing resources to the point that some considered their colleagues as family.

'Fishermen men are people who travel a lot in search of fish from one dock to another and if we move here to another dock, we usually make friends with other fishermen that side ….' FGD participant_F024_P1_23 years old

'... if your friend doesn't have food, we share with one another…. we meet fishermen from Blantyre or Zomba and when we meet in the boat, we act like children from the same family…. if it happens that one does not have any relative, we can say that 'my friend you will be staying with me'- FGD participant_F024_P3_30 years old

The webs of friendship and social connection seemed to transcend the immediate boat-team to include members of other boat-teams in their local and distant landing sites. Due to the unpredictable, physically demanding and occasionally dangerous nature of their occupation, participants cited that they relied on other boat teams if they faced any challenges both on-shore and off-shore. This practice was embedded in their culture and actively encouraged.

**Table 2. Summary of key findings.**

|  | Key findings | Selected Quotes |
|---|---|---|
| **Theme: Perceived significant social ties for fishermen** | | |
| Fishing team ties | Fishermen established important social ties with other fishermen who belonged to either their boat-team or other teams (both locally and at distant landing sites). These ties were important for knowledge exchange and resource-sharing. Mutual reliance was practiced and actively encouraged | *'Fishermen men are people who travel a lot in search of fish from one dock to another and if we move here to another dock, we usually make friends with other fishermen that side….'* <br> *FGD participant_F024_P1_23 years old* |
| Other social actors: kin, peers, chiefs and religious leaders | There was an overlap in the roles that social contacts played. Some dualled as work colleagues or as friends or relatives. In some areas, fishing teams were largely comprised of kin. Chiefs and religious leaders had a more global influence. They influenced the opinions of individual fishermen in various domains including health. | *'…. To be honest with you most of the fishermen here are related in one way or the other....'* <br> *IDI participant_SNI 016_33 years old* <br> *'…Chiefs are more important. They are the owners of the village and whatever happens in the communities goes through them first and they have powers of bringing us together….'* <br> *FGD participant_F024_P3_43 years old* |
| **Theme: Networks' influence on behavior and intervention uptake** | | |
| Knowledge exchange | Social ties were leveraged for sharing and acquiring knowledge about various matters including health. Uniquely, by nature of their profession, fishermen spend a significant time together and thus they share even sensitive matters related to health <br> The FISH trial introduced new knowledge and tools for self-care. The existing ties were used to diffuse the information | *'I would say that mostly, if there is one time where fishermen … tell one another about their secrets is during the time when they are fishing. …... In short, I would say that fishing is one of the jobs where people share different things that are important …'* <br> *FGD participant_F024_P8_27 years old* |
| Social Support, Trust and Conformity | Functional aspects of social networks such as social support and trust played a role in influencing the uptake of FISH trial interventions. Some participated by imitating those whom they trust, and some received encouragement and support to use self-testing kits. <br> Other participants joined the study due to the desire to align with peers and authority figures out of motivation, fear or obedience. | *'… we were only 7 fishermen who accepted to join, some fishermen were afraid.…I was among the 7 fishermen who accepted to join and receive the treatment … in the study and when other fishermen saw that we were not experiencing any problems, they started approaching us and asked more about the study and due to this, more than 100 fishermen joined the study because of this.'* <br> *IDI participant_SNI 009_39 years old* |
| Negative Peer influence | In some cases, inaccurate information about the study was spread among the fishermen and hindered uptake and study participation. | *'… wrong information prevented fishermen from understanding the true intentions of the study …. Fishermen with that wrong information, …… were spreading fake rumors that it was not about bilharzia but blood collection. It was a very big challenge to convince fishermen…'-* <br> *IDI participant_SNI 002_50 years old* |
| **Theme: Other factors affecting uptake of health services and interventions** | | |
| Facilitators: Perceived susceptibility and Financial incentives | The perceived susceptibility to HIV and schistosomiasis as an occupational risk was reinforced through shared experiences and discussions. This motivated some to participate in the study. There was increased interest and participation in the study when some participants heard of the reimbursements that were being offered as part of the study | *'Distributing the study message to people (fishermen) was easy because some friends who already participated were receiving money, so some other people(fishermen) started showing up without giving them any information.'* <br> *IDI participant_SNI 007_48 years old* |
| Barriers: HIV-related stigma and prohibitive traditional and religious beliefs | Negative perceptions about HIV shaped the response that some participants had towards the HIV services that were being offered in the trial. <br> Certain religious and traditional groups actively discouraged their members from seeking any health service including the services being offered in FISH | *Like some churches, they don't allow their members to go to the hospital even with HIV, they tell church members that they will get healed without going to the hospital.'* <br> *FGD participant_F001_P10_35 years old* |

*'…Sometimes a fisherman from one boat can have strong friendships with fishermen from the same boat teams... and fishermen of other boat teams.'-FGD participant_F043_P1_53 years old*

*'... we have rules at the dock that if such a scenario [a boat runs out of fuel or boat breakdown] happens to your friend, you should not leave him alone …, we should share with him some [fuel]. Those are some of the rules that make fishermen love one another. If there is a strong wind and you don't help your friend, he can suffer or even die.'- FGD participant_F024_P4_24 years old*

*'... as a fisherman, if you don't have social relationships nothing will work. Because all of us here are fishermen, and we move a lot from village to village. ….. if there's no unity here on the shore, it'll all end here. But when we go into the water, everyone is a brother.' FGD participant _F001_P1_60 years old*

**Kinship ties.**  Various ties with relatives were mentioned by study participants including those with parents, siblings and children. Participants expressed the value of kin in health decisions and provision of support. They cited that family members were quick to notice illness, respond to health concerns and provide the necessary support to engage in care. Additionally, due to the trust shared with family members, it was easier to open up about any health challenges.

*'…. To be honest with you most of the fishermen here are related in one way or the other. So, I would say that it was somehow easy for me to deliver the information to them because some of them are my friends and others are my relatives.'- IDI participant_SNI 016_33 years old*

*'In this community when one falls sick, it is hard for some people to know that he is sick unless one is a brother, and they encourage and take him to the hospital….' FGD participant _F043_P5_51 years old*

**Friendship and peer ties.**  Interdependencies outside of fishing provided an avenue for shared interests and a platform for forming valuable relationships to fishermen. Participants reported that they formed friendships with others whom they went to church with, socialized with or spent time with for any reason. These relationships catered to their social needs but also served as channels for sharing health-related challenges and information.

*'…. When we go the places where people play Bawo (board game) because a lot of people gather at such places…we make new friends at this place with people who are fishermen or not….'- FGD participant_F045_P5_47 years old*

*'…some make friends at the places where people smoke marijuana, some establish social relationships through fishing… So, … there are different forms of social relationships, fishermen create social relationships among themselves and with other people who are not fishermen….'- FGD participant_F001_P1_60 years old*

There was an apparent overlap in the roles that social contacts played in the lives of individual fishermen. In some cases, boat team members dualled as friends or relatives.

**Other social actors.**  Participants described other social actors whose influence spanned across the entire social system and affected the decisions of all or part of the system. These actors were sometimes not part of one's immediate social network.

**Chiefs:** Chiefs were reported as gatekeepers of the communities and their influence far-reaching, encompassing areas such as economics, development, health and politics. They also had an influence in fishing activities through formulation of by-laws which are enforced by the Beach Village Committee (BVC). Participants cited that chiefs acted as intermediaries between health workers and the community by approving interventions coming to their villages, mobilizing community members and ensuring that interventions are well-received.

*'…Chiefs are more important. They are the owners of the village and whatever happens in the communities goes through them first and they have powers of bringing us together….' - FGD participant_F024_P3_43 years old*

*'Chiefs, take a megaphone and announce to people that they must gather. … The chief is the one who speaks pertaining to the agenda of the gathering and instructs people to take part …. He may say "In this village we have cholera, so before the health personnel come, we should first meet as a village so that they find us prepared." - FGD_F001_P1_60 years old*

**Religious leaders:** Pastors and Sheikhs were described as people who provided moral and spiritual support to the community. Religious leaders were reportedly very influential to the fishermen as they are regarded to be chosen by God to serve people. Additionally, they are used as conduits of important messages to reach the community including health-related communications.

*'I listen to them because they're the leaders, even in the Bible they appointed leaders. You can't go somewhere where they're no leaders, so we have to listen to them because it's very important.'- IDI participant_SNI 004_59 years old*

*'In terms of sheikhs or at the church, they can take advantage of the days that people gather for prayers…. to inform people about health-related issues.' - IDI participant_SNI 016_33 years old*

**Networks' influence on behavior and intervention uptake**

**Knowledge exchange.** Participants cited leveraging their social networks to share or acquire knowledge on various matters including health. Despite their mobility, which can make consistent access to health information difficult, fishermen used informal communication to bridge these gaps. They mentioned that fishing activities create a conducive environment for sharing personal and communal information. Additionally, information sharing on health-related matters also occurred during their social interactions, such as at docks or drinking joints (bars).

*"... fishermen are wanderers which makes it difficult for some to receive messages regarding health issues. So, fishermen who come from areas where they receive messages regarding health issues …. and they go to another dock (where there is no access to health information). When they are chatting….in places such as beer drinking joints, they pass on the message to others …' FGD participant_F043_P1_69 years old*

*'When we received the information from the groups, we would also share it with other fishermen who didn't receive the information and that's how information spreads in the entire village.' IDI participant_SNI 004_59 years old*

*"I would say that mostly, if there is one time where fishermen … tell one another about their secrets is during the time when they are fishing. …... In short, I would say that fishing is one of the jobs where people share different things that are important …' FGD participant_F024_P8_27 years old*

Health interventions, such as the FISH trial, played a crucial role in introducing new knowledge and tools for self-care to fishermen.

*'When the FISH study came, we were called … and given self-test kits for HIV and its instructions about sample collection, testing and interpretation of results. I would say that prior to FISH study, I didn't know anything about self-testing for HIV, it was after the study when I realized that people can self-test for HIV.'- IDI participant_SNI 017_22 years old*

*'Many of us didn't know about the schistosomiasis prevention, but after the FISH study came, that's when we learned that it was dangerous to play in the lake.' – IDI participant_SNI 004_59 years old*

Subsequent to information provided by the trial personnel and peer educators, social relationships created windows of opportunity for the diffusion necessary for information to spread to other fishermen and the rest of the community.

*'Fishermen who were shy or who thought the beach clinics were far were motivated by their fellow fishermen who understood the study, … so the social relationships between fishermen really simplified the job.....'*- IDI participant_SNI 002_50 years old

Participants stated that information was mostly shared in-person and rarely was digital technology such as social media used. Participants narrated how information was shared in the FISH trial.

*'We used phone calls sometimes to reach out to those fishermen we know, but the information was delivered mainly one on one and in groups since we do our work in a team.'*- IDI participant_SNI 004_59 years old

**Social support.** Participants highlighted the importance of individualized support from social contacts that enabled them to participate in health activities. In the context of the FISH trial, some individuals preferred private interactions where support for HIV self-testing was provided. Additionally, peer relationships were reported as crucial in fostering emotional and material support.

*'... people who were willing to get the services that were being provided in the study [FISH Trial], were approaching me privately, so I could give information to these fishermen on a one-to-one basis….'* -IDI participant_SNI 016_33 years old

*'We are able to disclose to our friends how we are feeling in our bodies, and some give you advice that you should go to the hospital and in a situation where you don't have money, some assist you with money for transport (to go to the hospital) where one gets the treatment.'*- FGD participant_F001_P12_28 years old

**Trust and conformity.** Participants in both the FGDs and IDIs, concurred that familiarity and shared community ties made it easier to communicate and spread important health messages. Some explained that social relationships provided a supportive environment for individuals to discuss sensitive health concerns and encouraged timely health-seeking behaviors.

*'…. When one of us has a health problem, we are open to one another …. and in such situations, we try to encourage him that he should go to the hospital so that he can get treatment.'* -FGD participant_F001_P12_28 years old

*'I would say that it was easy because most of the fishermen who accepted the invitation to participate in the study were my friends as well as those whom I interact with on a regular basis at this dock and some of them were my relatives. …..'* IDI participant_SNI 016_33 years old

Additionally, several FISH trial participants recalled that shared bonds of friendship, family, and community created an environment where health-related messages were more likely to be positively received and acted upon by an individual. Observing others whom they trust, some participants who were skeptical about the study and its interventions were encouraged to participate.

*'… we were only 7 fishermen who accepted to join, some fishermen were afraid….I was among the 7 fishermen who accepted to join and receive the treatment … in the study and when other fishermen saw that we were not experiencing any problems, they started approaching us and asked more about the study and due to this, more than 100 fishermen joined the study because of this.'* IDI participant_SNI 009_39 years old

*'Aaaah... it was very hard for people to accept the study but there are some people who are close to you who listen to you whenever you tell them something, they understand you very easily, I would say that some of my friends accepted to participate in the study ....' IDI participant_SNI 017_22 years old*

Participants also cited that the desire to align with peers or authority figures motivated them to engage with health interventions or services in some cases. In the FISH trial, participants noted that some individuals opted to accept treatment after observing others in their social circles doing the same. Others participated out of fear or a sign of obedience to those in authority

*'…When the fishermen saw that their bosses participated in the study and received the treatment for HIV or Schistosomiasis from the study staff, they thought that it would be good for them as well to accept the treatment, and they followed the same path….' IDI participant_SNI 016_33 years old*

**Peer influence.** Various factors such as trust, knowledge exchange and desire to conform acted together to bring about positive peer influence. This influence was shown to promote good health-seeking behavior among fishermen and to actively engage with the FISH trial interventions. Participants highlighted how they openly discuss health issues within their social circles, seeking advice and encouraging one another to visit health facilities. Additionally, individuals who had previously received treatment as part of the FISH trial and experienced its benefits often shared their positive experiences with others, motivating their peers to seek similar care and follow recommended guidelines.

*'... fishermen are open to one another when we are sick. We tell our friends that 'my friend, I am not feeling ok and what should I do?' and we are able to advise such fishermen that they should go to the hospital or visit the HSAs …and people are able to go to the hospital.' FGD participant _F043_P11_20 years old*

*'… so if someone had done it or received the treatment before and they received help, they would also tell their friends about its benefit, that if you want it to work, you should do this.' IDI participant_SNI 007_48 years old*

*'I would say that most of the people when they are in groups let's say of ten with the same problem and if one of them goes to seek assistance at the clinic, the remaining 9 wait to see what happens to their friend... When they see that nothing bad happened to their friend, they are also motivated to go and get the treatment ….' IDI_SNI 019_39 years old*

**Misinformation.** Misinformation about the FISH study, particularly regarding its objectives and procedures, emerged as a key barrier to participation. The role of social networks in disseminating inaccurate information was evident in the way rumors spread among fishermen. For instance, some participants believed that the study's blood collection process was linked to strange government agendas rather than its stated purpose of testing for HIV.

*'… wrong information prevented fishermen from understanding the true intentions of the study …. Fishermen with that wrong information, …… were spreading fake rumors that it was not about bilharzia but blood collection. It was a very big challenge to convince fishermen and let them know the right information and the true intentions of the FISH Study, but later they realized that it was important…'- IDI participant_SNI 002_50 years old*

*'... some fishermen would discourage their friends from receiving the treatment that when you go there and get the treatment, they are going to pump and drain blood from your body and some were discouraging their friends that when you go there, they are going to test you for HIV, and these brought a lot of fears among some fishermen.' IDI participant_SNI 016_33 years old*

## Other factors affecting the uptake of health services and interventions

Participants in both FGDs and IDIs cited well known and documented factors operating at different levels- structural, community, individual and economic- that interacted with social networks to either hinder or facilitate the uptake of health interventions. Notably, factors such as stigma and traditional beliefs featured highly in the discussions and interviews.

**Facilitators. Perceived susceptibility:** Beyond the influence of others in the social environment, there was a general perception that fishermen have a heightened vulnerability to certain diseases such as HIV and schistosomiasis by nature of their occupation. This perceived susceptibility was reinforced through social networks, where shared experiences and discussions about health risks influenced decision-making. These dynamics were evident in their engagement with the FISH trial, which specifically targeted these occupational health challenges.

*'…I would say that fishermen consider themselves as being at risk of schistosomiasis because they are usually found mostly on the lake so they accepted the treatment from the FISH study so that they should be protected …' IDI participant_SNI 016_33 years old*

*'What motivated me to go and get tested was that we (fishermen) move around in different areas of which it is easy to get infected, so it turns out to be an opportunity for us if health providers came to our area to deliver their services for free…' IDI participant_SNI 013_45 years old*

**Financial incentives:** Financial reimbursements are a key feature of study participation, and they played a role in motivating participation in the FISH trial among the fishermen. These financial rewards had a ripple effect, where word-of-mouth spread among peers led to increased interest and participation. Moreover, the financial support helped participants to have funds to meet their families' daily needs, further reinforcing their engagement in the trial.

*'Distributing the study message to people (fishermen) was easy because some friends who already participated were receiving money, so some other people(fishermen) started showing up without giving them any information.' IDI participant_SNI 007_48 years old*

*'Some were motivated to participate in the study because of the money that they were being given after participating in the study.' -IDI_SNI 016_33 years old*

**Barriers. HIV-related social stigma:** Stigma emerged as a barrier to seeking health services among the participants, particularly in the context of HIV-related services. Negative perceptions about HIV and antiretroviral therapy (ART) contributed to fear and discrimination, shaping how individuals approached their health needs.

*'When people see someone going to the hospital to take the ARVs (Antiretroviral therapy), they judge him/her as someone who was engaging in promiscuous behaviors and people say a lot of negative things about someone who is HIV positive'. IDI participant_SNI 016_33 years old*

*'…. people who are living with HIV are stigmatized since most people think that such people can transmit the virus to others anyhow.' IDI participant _SNI 019_39 years old*

Stigma also shaped the response that some fishermen had towards the HIV services that were being provided by the FISH trial. They did not openly engage with the services as they feared the reaction of their peers.

*'I would say that most.. of them are involved promiscuous acts with women along the lake shore and … when they heard that people have come here with assistance for HIV and schistosomiasis, they were shy to come and ask more about the services … …' IDI participant_SNI 016_33 years old*

**Traditional and religious beliefs:** The diverse religious and tribal backgrounds in Mangochi played a role in shaping health-seeking behaviors, influencing whether individuals accepted or refused medical treatment or study participation. Within these social networks, certain religious groups actively discouraged their members from seeking care at health facilities, reinforcing mistrust in conventional medicine. This collective influence extended beyond individual beliefs, as refusal to seek treatment was often reinforced through peer pressure and shared values within these religious groupings.

*In this community there are a lot of people, some hold beliefs from their places of origin, ….. There are some people here who don't take drugs when they are sick…. Sometimes we offer them money to go and buy medication, but they refuse, they say that they don't take hospital medication. ….' FGD participant_F024_P4_24 years old*

*'Like some churches, they don't allow their members to go to the hospital even with HIV, they tell church members that they will get healed without going to the hospital.' FGD participant_F001_P10_35 years old*

## Discussion

This study explored the social networks and dynamics influencing the health decisions of fishermen in Mangochi, Malawi, particularly in the context of HIV and schistosomiasis interventions. Findings revealed that health decisions of fishermen are deeply embedded in their social structures. However, other factors operating at the individual and structural level were noted to be crucial catalysts to the decision-making process. This study, to our knowledge, is the first to qualitatively explore the influence of social networks in mediating the adoption of integrated interventions in a fishing population.

The influence of social networks in health intervention uptake has been documented in diverse populations and targeting various conditions [7,8,34–36]. However, most studies are quantitative in nature and fail to highlight the mechanisms of this influence. In this study we sought to understand from recipients of the FISH trial intervention how social interactions enhanced or hindered the uptake of the interventions and other factors necessary to the health decision-making process. We found that, given the communal and mobile nature of fishing activities, boat team members spent prolonged time periods together, which in turn cultivated the development of unique friendship bonds amongst boat crew members but also with members of other boats. Such strong intra- and inter-boat friendship bonds created unique windows of opportunity for fishermen to frequently share personal experiences and openly discuss sensitive topics, such as HIV testing and treatment. Evidence supports that inter-boat ties are strategic and that fishermen draw on these relationships to acquire information on fishing opportunities and sharing of resources [21,24,37]. It was interesting to observe health information being relayed through these ties despite being initially established primarily for other purposes. Perhaps researchers can leverage these occupational ties and established social bonds to serve as platforms for knowledge exchange and mutual support for targeted health interventions. There is empirical evidence to demonstrate that the increase in face-to-face interactions, as occurs in fishing, encourages the exchange of ideas [38]. Social ties transcended the fishing teams to encompass family, friends and community leaders, thus creating additional avenues for the diffusion of health information and support. These overlapping relationships enabled a broader reach of the FISH trial.

The FISH trial targeted fishermen who are traditionally at a heightened risk of HIV and schistosomiasis due to the nature of their work and are not adequately engaged in formal health services due to their substantial mobility [14]. The positive effects observed in the uptake of the trial interventions demonstrate that networks were instrumental in bridging informational gaps and catalyzing wider acceptance of the interventions [28]. Peer influence stemming from trust, conformity and compliance ensured that the message regarding the interventions was easier to share and to be acted upon. There is evidence to support the positive influence that social contacts bring to enhance interventions targeting HIV testing and men [39,40].

The effectiveness of social networks in promoting the uptake of interventions was countered by several factors operating at the individual, network and structural level. In some instances, the same social fabric that enhanced uptake

also facilitated the spread of negative influence and misinformation, which hindered uptake. This observation has been demonstrated in multiple studies aimed at increasing the uptake of various interventions, including vaccines, through both in-person and online approaches [41–44]. Participants also cited known barriers to adoption of HIV and schistosomiasis services such as stigma and prohibitive traditional and religious beliefs. This is consistent with other studies conducted in the region [25,43,45]. Thus, interventions designed to use social networks to accelerate the uptake of interventions should incorporate efforts to counter misinformation and address known barriers that hinder engagement with the targeted health services. For example, pre-intervention education campaigns done in collaboration with trusted community leaders can be used to sensitize communities on anticipated barriers and address misinformation. Additionally, digital platforms can also be used to disseminate culturally tailored content and counter any negative rumors that can arise.

The findings of this study make an important contribution to the literature on the influence of social networks in the uptake of integrated targeted health interventions. Nevertheless, the findings should be interpreted in light of several limitations. First, the study was conducted a year after the trial was concluded, potentially affecting the participants' ability to recall their experiences during the trial. However, since these communities are not regularly engaged in study activities, during data collection we observed that their recollection was not significantly impaired. Second, it was challenging to qualitatively understand who is connected to whom and define information flow trajectories within the network. Complementary quantitative methods are required to map the networks to visualize the social structure and analyze patterns of interaction, influence, and knowledge exchange. This approach can provide deeper insights into how relationships facilitate or hinder the spread of information and resources. Lastly, the study was conducted in one district only, limiting the generalizability to other districts with different socio-cultural, economic, or healthcare contexts. Future studies should consider recruiting from multiple contexts to enhance broader applicability.

## Conclusion

This study highlights the importance of leveraging social networks in the design and implementation of health interventions. However, peer networks, while powerful tools for information sharing, can also perpetuate resistance to health services emphasizing the need for strategies that also counter misinformation. Strategies should equally integrate culturally sensitive approaches that acknowledge traditional and religious beliefs to further enhance their effectiveness.

## Acknowledgments

We thank all study participants, peer leaders, and community leaders of the communities that participated in the study. We also thank Rowland Meja, Kondwani Phiri, Wisdom Sosola and Kennedy Mwema for their unwavering dedication and tireless efforts during data collection.

## Author contributions

**Conceptualization:** Madalo Mukoka, Guy Harling, Alison Price, Katherine Fielding, Augustine T Choko, Moses K Kumwenda.

**Data curation:** Madalo Mukoka, Owen Mhango, Moses K Kumwenda.

**Formal analysis:** Madalo Mukoka, Owen Mhango, Chikumbutso Chipandwe, Moses K Kumwenda.

**Funding acquisition:** Marriott Nliwasa, Chisomo Msefula, Augustine T Choko.

**Investigation:** Madalo Mukoka, Owen Mhango, Hussein H Twabi, Robina Semphere, Takondwa C Msosa, Marriott Nliwasa, Chisomo Msefula, Guy Harling, Alison Price, Katherine Fielding, Augustine T Choko, Moses K Kumwenda.

**Methodology:** Madalo Mukoka, Augustine T Choko.

**Project administration:** Madalo Mukoka, Owen Mhango, Chikumbutso Chipandwe.

**Writing – original draft:** Madalo Mukoka.

**Writing – review & editing:** Madalo Mukoka, Owen Mhango, Hussein H Twabi, Chikumbutso Chipandwe, Robina Semphere, Takondwa C Msosa, Marriott Nliwasa, Chisomo Msefula, Guy Harling, Alison Price, Katherine Fielding, Augustine T Choko, Moses K Kumwenda.

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
