## [Decision Letter · Decision Letter 0]

PGPH-D-25-00450

The influence of social networks in adoption of integrated health interventions: a qualitative study of fishermen in Malawi

Dear Dr. Mukoka,

Thank you for submitting your manuscript to PLOS Global Public Health. After careful consideration, we feel that it has merit but does not fully meet PLOS Global Public Health’s publication criteria as it currently stands. Therefore, we invite you to submit a revised version of the manuscript that addresses the points raised during the review process.

Pay attention to the points that all three reviewers have raised to strengthen the respective aspects of the manuscript they are indicated.

We look forward to receiving your revised manuscript.

Kind regards,

Ferdinand C Mukumbang, PhD

Academic Editor

Journal Requirements:

Additional Editor Comments (if provided):

Reviewers' comments:

Reviewer's Responses to Questions

**Comments to the Author**

1. Does this manuscript meet PLOS Global Public Health’s publication criteria ? Is the manuscript technically sound, and do the data support the conclusions? The manuscript must describe methodologically and ethically rigorous research with conclusions that are appropriately drawn based on the data presented.

Reviewer #1: Yes

Reviewer #2: Yes

Reviewer #3: Yes

2. Has the statistical analysis been performed appropriately and rigorously?

Reviewer #1: N/A

Reviewer #2: N/A

Reviewer #3: Yes

3. Have the authors made all data underlying the findings in their manuscript fully available (please refer to the Data Availability Statement at the start of the manuscript PDF file)?

Reviewer #1: Yes

Reviewer #2: No

Reviewer #3: Yes

4. Is the manuscript presented in an intelligible fashion and written in standard English?

Reviewer #1: Yes

Reviewer #2: Yes

Reviewer #3: Yes

5. Review Comments to the Author

Reviewer #1: A great work highlighting the utility of social networks in influencing health decisions.

Your manuscript would benefit, especially in the discussion section, from a section that discusses current limitations in how social networks are used in health interventions and how your recommendations mitigate these limitations. A great way to start was when you stated that social network interventions should incorporate mechanisms to mitigate the spread of misinformation. You could for instance include recommendations on what these mechanisms could be. And then just as this recommendation directly addresses the limitation of misinformation in social networks, you can tailor your other recommendations to do address other limitations.

Reviewer #2: A well-conducted qualitative study examining the influence of social networks on Malawian fishermen's adoption of integrated health interventions is presented in this publication. For public health initiatives aimed at high-risk and mobile populations, the study is extremely significant. However, there is room for improvement in a few areas.

1. I am still unsure how social networks influence health behavior through mechanisms like social norms, social capital, and peer influence. Perhaps, a conceptual model or diagram illustrating how different levels of social networks impact health intervention adoption would be beneficial.

2. A more clearer description of the coding framework used would help to refine the data analysis part.

3. A table of the important key findings with themes, sub-themes and quotes would look more condensed and easy to read and understand.

4. I would suggest compressing few quotes; some are too lengthy.

5. Since the study was conducted in a single district, acknowledging this as geographic limitation would be useful.

Reviewer #3: Codebook Development: Please specify the total number of codes generated and provide a detailed description of the process used to create the codebook. Clarify as well the role of the senior researcher in this process. It would also be helpful to indicate which aspects of coding were influenced by a deductive versus an inductive approach.

6. PLOS authors have the option to publish the peer review history of their article (what does this mean? ). If published, this will include your full peer review and any attached files.

**Do you want your identity to be public for this peer review?** For information about this choice, including consent withdrawal, please see our Privacy Policy .

Reviewer #1: No

Reviewer #2: No

Reviewer #3: **Yes: ** Angele Bienvenue Ishimwe

---

## [Editor Report · Decision Letter 1]

The influence of social networks in adoption of integrated health interventions: a qualitative study of fishermen in Malawi

PGPH-D-25-00450R1

Dear Dr Mukoka, 

We are pleased to inform you that your manuscript 'The influence of social networks in adoption of integrated health interventions: a qualitative study of fishermen in Malawi' has been provisionally accepted for publication in PLOS Global Public Health.

Best regards,

Ferdinand C Mukumbang, PhD

Academic Editor